# Preeclampsia: Risk Factors, Diagnosis, Management, and the Cardiovascular Impact on the Offspring

**DOI:** 10.3390/jcm8101625

**Published:** 2019-10-04

**Authors:** Rachael Fox, Jamie Kitt, Paul Leeson, Christina Y.L. Aye, Adam J. Lewandowski

**Affiliations:** 1Oxford Cardiovascular Clinical Research Facility, Division of Cardiovascular Medicine, Radcliffe Department of Medicine, University of Oxford, Oxford OX3 9DU, UK; 2University of Melbourne, Victoria 3010, Australia; 3Nuffield Department of Women’s and Reproductive Health, University of Oxford, Oxford OX3 9DU, UK

**Keywords:** foetus, preeclampsia, pregnancy, foetal diseases, prevention, treatment, developmental origins of disease, non-communicable disease

## Abstract

Hypertensive disorders of pregnancy affect up to 10% of pregnancies worldwide, which includes the 3%–5% of all pregnancies complicated by preeclampsia. Preeclampsia is defined as new onset hypertension after 20 weeks’ gestation with evidence of maternal organ or uteroplacental dysfunction or proteinuria. Despite its prevalence, the risk factors that have been identified lack accuracy in predicting its onset and preventative therapies only moderately reduce a woman’s risk of preeclampsia. Preeclampsia is a major cause of maternal morbidity and is associated with adverse foetal outcomes including intra-uterine growth restriction, preterm birth, placental abruption, foetal distress, and foetal death in utero. At present, national guidelines for foetal surveillance in preeclamptic pregnancies are inconsistent, due to a lack of evidence detailing the most appropriate assessment modalities as well as the timing and frequency at which assessments should be conducted. Current management of the foetus in preeclampsia involves timely delivery and prevention of adverse effects of prematurity with antenatal corticosteroids and/or magnesium sulphate depending on gestation. Alongside the risks to the foetus during pregnancy, there is also growing evidence that preeclampsia has long-term adverse effects on the offspring. In particular, preeclampsia has been associated with cardiovascular sequelae in the offspring including hypertension and altered vascular function.

## 1. Introduction

Hypertensive disorders of pregnancy affect 10% of pregnancies [1] and are defined by the International Society for the Study of Hypertension in Pregnancy (ISSHP) as new onset hypertension (≥140 mmHg systolic or ≥90 mmHg diastolic) after 20 weeks’ gestation [2]. This umbrella definition includes chronic hypertension, gestational hypertension and preeclampsia (de novo or superimposed on chronic hypertension). Both of these conditions can have significant impacts on maternal and foetal health in the immediate and long term. For the mother, this includes a two- to four-fold increased risk of long-term hypertension, a doubling of the risk of cardiovascular mortality and major adverse cardiovascular events, and a 1.5-fold increased risk of stroke [3]. For the foetus, this includes antenatal risks of intra-uterine growth restriction (IUGR), preterm birth (most commonly iatrogenic), oligohydramnios, placental abruption, foetal distress, and foetal death in utero [4,5,6]. There is also growing evidence that in utero exposure to hypertensive disorders of pregnancy can result in significant long-term cardiovascular sequelae in the offspring, including early onset hypertension, and an increased risk of ischemic heart disease and stroke [7]. These sequelae have been associated with hypertensive pregnancies independent of other coexisting pregnancy complications. This article reviews the latest evidence base and guideline updates surrounding the diagnosis, management, and foetal surveillance in preeclampsia, as well as its increasingly recognised role as an independent cardiovascular risk factor for the offspring.

## 2. Risk Factors for Preeclampsia and Risk Reduction

The 2019 National Institute for Health and Care Excellence (NICE) guidelines [3] classify a woman at high risk of preeclampsia if there is a history of hypertensive disease during a previous pregnancy or a maternal disease including chronic kidney disease, autoimmune diseases, diabetes, or chronic hypertension. Women are at moderate risk if they are nulliparous, ≥40 years of age, have a body mass index (BMI) ≥ 35 kg/m [2], a family history of preeclampsia, a multifoetal pregnancy, or a pregnancy interval of more than 10 years [3]. These risk factors are echoed in the largest meta-analysis of clinical risk factors to date conducted by Bartsch et al. [8], who analysed over 25 million pregnancies from 92 studies. The presence of one high risk factor, or two or more moderate risk factors, is used to help guide aspirin prophylaxis, which is effective in reducing the risk of preeclampsia if administered before 16 weeks of pregnancy [9,10].

There are additional clinical factors that significantly increase preeclampsia risk, including raised mean arterial blood pressure before 15 weeks’ gestation [11], polycystic ovarian syndrome [12,13,14], sleep disordered breathing [15], and various infections such as periodontal disease, urinary tract infections [16], and helicobacter pylori [17,18]. In terms of obstetric history, vaginal bleeding for at least five days during pregnancy increases preeclampsia risk [11], as does the use of oocyte donation, which has a higher risk of preeclampsia in comparison to in vitro fertilization (IVF) without oocyte donation or natural conception [19,20,21].

Biochemical and ultrasound markers are being investigated as additional predictors for preeclampsia. Foetal factors including genotype and foetal cell-free DNA in maternal blood can influence a woman’s risk of preeclampsia [22,23]. Recently, a genome-wide association study of 4380 cases of preeclampsia and 310,238 controls identified that a variant in the foetal genome near the locus of fms-like tyrosine kinase-1 (Flt-1) is implicated in the development of preeclampsia [22]. Increased cell-free foetal DNA in maternal blood is another potential marker, and is detectable before onset of symptoms [24]. The most promising foetal and placental biomarkers for identifying preeclampsia are placental growth factor (PlGF) and soluble Flt-1 (sFlt-1), which are discussed subsequently. Meta-analyses have described a potential association between preeclampsia and elevated levels of serum triglycerides, cholesterol, and inflammatory markers including CRP, IL-6, IL-8, and TNFα, some of which precede the onset of preeclampsia [25,26,27,28]. Uterine artery Doppler analysis has mixed results in predicting preeclampsia [11,29,30]. A recent meta-analysis reported that use between 11 and 14 weeks can predict preeclampsia with similar accuracy as clinical risk factors [30]. Incorporation of specialist tests such as uterine artery pulsatility index and pregnancy-associate plasma protein A (PAPP-A) into clinical risk prediction models can also increase the positive predictive value for detecting women at risk of this condition [31].

At present, aspirin is the only therapy with robust evidence supporting its use to reduce the risk of preeclampsia in high-risk women [32]. Current recommendations advise low dose (75–150 mg) aspirin as prophylaxis from 12 weeks’ gestation until delivery [3]. When taken before 16 weeks’ gestation, low dose aspirin has a modest, but consistent effect, estimated to reduce the risk of preeclampsia by approximately 10% [9]. Other interventions including nutritional supplements, pharmacological agents, and dietary and lifestyle interventions have been investigated for protective effects against preeclampsia with varying efficacy. Studies have reported that vitamin D deficiency can increase the risk of preeclampsia [33,34,35,36,37,38], and that vitamin D supplementation may offer some benefit in reducing preeclampsia risk [38,39]. However, while supplementation is often recommended in clinical practice, robust randomised controlled trial (RCT) evidence is still required to confirm its utility [37,38,40]. The World Health Organization conducted a large RCT investigating the role of calcium, reporting no decrease in preeclampsia with supplementation in a calcium-deficient population, although the severity and complications of preeclampsia were significantly lower in the supplementation cohort [41]. A 2018 Cochrane review of high dose (>1 g/day) calcium supplementation from 20 weeks’ gestation did see a reduction in preeclampsia, although this finding relied on small studies and is likely an over-estimate [42]. Current guidelines cite this as evidence for calcium supplementation in deficient pregnant women [43,44]. Supplementation of the antioxidants vitamins C and E has no benefit in preventing preeclampsia [45,46,47,48] despite initial promising results [49]. Likewise, high dose folic acid does not appear to have any preventative effects [50], though some evidence suggests that supplementation with 5-methyl-tetrahydrofolate supplementation, a more bioavailable form of folic acid, may be effective in preventing recurrent preeclampsia [51].

A Cochrane review reported that antithrombotic agents such as low molecular weight heparin reduce preeclampsia risk in women at higher risk of placental insufficiency, but significant heterogeneity between the studies limited the certainty of conclusions [52]. A variety of other pharmaceutical agents have been assessed and in small studies. L-arginine, pravastatin, coenzyme Q10, and ketanserin have all been associated with lower rates of preeclampsia. However, larger studies are needed to evaluate their efficacy and safety [53,54,55,56].

Lifestyle interventions may reduce preeclampsia, particularly dietary interventions [57,58]. Lower rates of preeclampsia may be associated with a higher vegetable and plant-based diet [11,59], however, findings are mixed. A recent RCT reported that antenatal lifestyle advice (diet and exercise) had no effect on preeclampsia rates in overweight or obese women [60]. Exercise interventions seem to have a more limited effect than diet [61], although there is insufficient evidence to make consistent conclusions [62]. Surprisingly, smoking has been identified as a protective factor against the development of preeclampsia, with risk inversely proportional to number of cigarettes smoked [63].

## 3. Diagnosis

Internationally, preeclampsia is defined as new-onset gestational hypertension (systolic blood pressure ≥140 mmHg and/or diastolic blood pressure ≥90 mmHg) associated with new-onset of at least one of proteinuria, maternal organ dysfunction (liver, neurological, haematological, or renal involvement), or uteroplacental dysfunction at or after 20 weeks’ gestation (Table 1) [2]. It is important to note that preeclampsia may develop for the first time intrapartum or postpartum. Super-imposed preeclampsia can also be diagnosed in women with chronic hypertension who develop new onset proteinuria, maternal organ, or uteroplacental dysfunction consistent with preeclampsia [2]. Eclampsia occurs when there are convulsions in the setting of preeclampsia [3].

### 3.1. Blood Pressure

In order to confirm the presence of hypertension, blood pressure should be measured on at least two occasions four hours apart using an appropriately sized cuff and validated device for use in women with preeclampsia [2,64]. For women at high risk, guidelines recommend monitoring blood pressure at increased frequency in antenatal clinics, however no exact frequency is recommended. Recent studies have addressed the potential for women to self-monitor their blood pressure at home to improve the detection of hypertension in pregnancy, particularly in women with elevated risk. It appears that self-monitoring is feasible [65,66], acceptable to pregnant women [67], may reduce clinic visits [66], and be effective for detecting hypertension in pregnancy and distinguishing white coat hypertension [68]. A current RCT (BUMP) hopes to provide a larger evidence base to determine the impact of self-monitoring on maternal and neonatal outcomes and advise how self-monitoring can be implemented into clinical practice (www.phc.ox.ac.uk/research/participate/bump-trial).

### 3.2. Proteinuria

The presence of proteinuria has traditionally been screened for by dipstick testing and confirmed by additional laboratory tests using 24 h urine, or more recently spot samples of urine. Screening assessment with dipstick testing is best done with an automated reagent-strip reading device rather than visual analysis [69]. Previously, 24 h urine collection was considered the gold standard for confirmation of proteinuria but it has several problems: it is time consuming, requires refrigeration, samples are often incomplete, and is infrequently used in hospitals [70]. Therefore, after a positive dipstick test (one protein or more), the use of either spot urine albumin to creatinine (A:Cr) or protein to creatinine (P:Cr) ratios are now recommended to quantify proteinuria [3]. Both P:Cr and A:Cr testing are shown to significantly correlate with proteinuria as detected by 24 h urine [71,72,73,74,75]. Diagnostic thresholds of 30 mg/mmol and 8 mg/mmol have been determined to provide high sensitivity and specificity, respectively [71,73].

### 3.3. Laboratory and Imaging Tests

The ISSHP recommends that pregnant women with de novo hypertension are investigated with laboratory tests measuring haemoglobin, platelet count, serum creatinine, liver enzymes, and serum uric acid to determine the presence of maternal organ dysfunction and the diagnosis of preeclampsia [2]. New guidelines have also implemented PlGF or sFlt-1:PlGF ratio testing for preeclampsia diagnosis in specific circumstances [3]. There is a large body of work indicating a role of circulating angiogenic factors, such as sFlt-1 and PlGF, in the pathogenesis of preeclampsia. Women with preeclampsia have higher circulating levels of sFlt-1 and lower levels of PlGF, noticeable before the onset of the disease [76]. SFlt-1 is an anti-angiogenic protein that acts as an antagonist to the angiogenic proteins PlGF and vascular endothelial growth factor (VEGF). By inhibiting VEGF and PlGF, sFlt-1 alters downstream signalling pathways, which results in vasoconstriction and endothelial dysfunction [77]. Increasing sFlt-1 levels in mouse models have been shown to produce a syndrome resembling preeclampsia. Furthermore, removing sFlt-1 can reverse endothelial dysfunction in endothelial cell culture studies; hence overexpression appears an important mechanistic link between placental dysfunction and altered maternal vascular function [76]. Low PlGF has been shown to have a high sensitivity and negative predictive value in diagnosing preeclampsia needing delivery within 14 days [78]. A large UK based stepped-wedge cluster-randomised controlled trial showed that those who had revealed PlGF testing received a diagnosis of preeclampsia significantly faster, with a significant reduction in adverse maternal events and no change in neonatal adverse outcomes [79]. Another large trial demonstrated that a sFlt-1:PlGF ratio of <38 can rule out preeclampsia within the next seven days [80]. NICE have adopted this research, recommending the use of PlGF or sFlt-1:PlGF ratio to help rule out preeclampsia in women between 20 and 34 + 6 weeks’ gestation in whom preeclampsia is suspected. It is not currently recommended to rule in preeclampsia [3].

Uteroplacental dysfunction can be evaluated with ultrasound assessment of foetal growth and umbilical artery Doppler velocimetry or cerebroplacental ratio measurements to assess blood flow redistribution in placental insufficiency [3].

## 4. Impact on the Foetus

### 4.1. Outcomes

The pathogenesis of preeclampsia is complex and not fully understood, however it is known to involve dysfunctional placentation, systemic inflammation, and oxidative stress [81] as has been described elsewhere [81,82,83]. Abnormal placentation occurs due to failure of appropriate remodelling of the spiral arteries, resulting in higher resistance to placental blood flow and hypoperfusion of the placenta. This causes chronic placental ischemia and reduced blood flow to the developing foetus [81]. These maladaptive processes can precipitate foetal hypoxia and adverse outcomes including IUGR, preterm birth (both spontaneous and iatrogenic), oligohydramnios, placental abruption, foetal distress, and foetal death in utero [4,5,6]. The frequency of foetal complications differs depending on the onset of preeclampsia. Early onset of preeclampsia has been associated with significantly higher rates of adverse outcomes for the foetus, including IUGR, oligohydramnios, and foetal death [4,5].

### 4.2. Surveillance and Diagnosis of Complications

Currently, there is no established consensus regarding the optimal monitoring of the foetus in preeclamptic pregnancies. Guidelines developed by the UK, USA, Canada, Australia, and New Zealand differ significantly in the modalities recommended for foetal surveillance and the frequency of assessment they recommend, as illustrated in Table 2. In general, measures used for foetal surveillance include maternal reports of foetal movements, biophysical profile (BPP), cardiotocography (CTG), amniotic fluid volume (AFV) assessment, ultrasound assessment of foetal growth, as well as ultrasound Doppler measurements of the umbilical artery, ductus venosus, middle cerebral artery, and cerebroplacental ratio.

Pregnant women with preeclampsia are often encouraged to monitor foetal movements and report any changes to a health practitioner [44]. However, as a surveillance technique, daily monitoring of foetal movements does not appear to improve foetal outcomes or to prevent stillbirth [85]. BPP testing, which utilizes antenatal CTG alongside ultrasound assessments of foetal movements, breathing, tone, and amniotic fluid volume, is another method of monitoring foetal wellbeing, and recommended in American guidelines [44]. However, use of BPP is not supported by current evidence in high risk pregnancies, including those complicated by preeclampsia [86]. Australian and US guidelines recommend antenatal CTG to measure foetal heart rate at diagnosis of preeclampsia, and subsequently twice weekly for assessment of foetal wellbeing [43,44]. Nevertheless, there is no clear evidence that antenatal CTG monitoring improves perinatal outcomes [87]. As a result, the latest NICE guidelines state to repeat CTG ‘when clinically indicated’ rather than routinely [3]. Other modalities for foetal assessment include ultrasound measurement of foetal weight, which is used at diagnosis of preeclampsia, and then two to three times weekly with the aim to detect IUGR [3,43]. Assessment of AFV allows the detection of oligohydramnios and is also recommended as part of foetal assessment [3,43]. AFV can be assessed with ultrasound measurements of the single deepest vertical pocket or by amniotic fluid index, with similar prevention of perinatal outcomes [88].

Unlike previously mentioned investigations, the use of umbilical artery Doppler ultrasound for foetal surveillance in high risk pregnancies is supported by several RCTs and systematic reviews [89,90,91]. Doppler ultrasound of the foetal umbilical artery measures blood flow patterns through the artery as an indicator of placental perfusion. If the umbilical artery flow is absent or reversed during end-diastole, this illustrates abnormally high placental resistance, hence reduced foetal blood flow, and is an indicator of risk of adverse foetal outcome including perinatal mortality [92]. Additionally, if a foetus is compromised, Doppler ultrasound changes precede changes in foetal heart rate, allowing earlier detection [93]. The majority of evidence regarding the use of umbilical artery Doppler comes from trials of ‘high risk’ pregnancies, in which preeclamptic pregnancies are included. However, there are very limited studies addressing the use of umbilical artery Doppler in preeclamptic pregnancies specifically. One of the largest RCTs involving umbilical artery Doppler assessments includes 1340 women with high risk pregnancies randomized into surveillance with CTG testing or umbilical artery Doppler velocimetry [89]. They reported significantly lower rates of caesarean sections for foetal distress in the cohort undergoing Doppler assessments. Notably, this impact was most pronounced in women with high risk pregnancies due to maternal hypertension. The reduction of emergency caesareans was suggested to be a result of earlier detection of foetal distress, allowing enough placental reserve for the foetus to endure labour. An early systematic review of 12 RCTs evaluating the use of umbilical artery Doppler ultrasound in high risk pregnancies reported a 38% reduction in odds in perinatal death [91]. While this reduction in risk of perinatal death is compelling, it is thought to be overstated. A more recent Cochrane review reported a 29% reduction in mortality [90]. These mixed results are largely explained by the heterogeneous nature of studies due to differences in study design. This includes different classifications of high risk pregnancies, gestational age at inclusion, institutional procedures, Doppler velocimetry equipment and settings, ultrasound operators, and the number of examinations, which limit the results from meta-analyses [94]. Regardless, re-examination of these RCTs has shown enough evidence to confirm umbilical artery Doppler velocimetry does reduce unnecessary obstetric interventions (such as induction of labour and caesarean sections) and perinatal mortality, particularly in pregnancies complicated by IUGR or preeclampsia, and should be conducted in these cases [94]. There remains a lack of evidence to guide specific recommendations regarding initial timing of surveillance with umbilical artery Doppler ultrasound and the ongoing frequency of its use. The studies analysed in the Cochrane review mentioned above differ in the frequency of Doppler assessments conducted and in the gestational age at which analysis was commenced [90]. Some conducted umbilical artery Doppler assessments multiple times per week, while others at fortnightly intervals. As absence or reversal of end diastolic flow is unlikely to occur within seven to 10 days of a normal umbilical artery Doppler analysis [43], in the absence of abnormal findings, guidelines often recommend repeat assessment fortnightly from diagnosis until birth [3,43]. Nevertheless, the optimal timing and frequency of umbilical artery Doppler assessment remain elusive.

Additional Doppler ultrasound measurements used in assessment include monitoring of the ductus venosus or the foetal middle cerebral artery (MCA). Changes in flow through the ductus venosus occur in placental compromise and hypoxia [95], and Doppler assessment can be useful in assessing foetal wellbeing in early onset placental dysfunction [96]. Resistance in the cerebral arteries decreases with worsening growth restriction and hypoxemia as a mechanism to prioritize blood flow to the brain [97]. An abnormally low pulsatility index (PI), a marker of vascular impedance, in the MCA has been observed in preeclamptic pregnancies including those complicated by IUGR, particularly in later gestations [97,98]. However, on its own the MCA is not necessarily a reliable indicator of poor foetal outcome [98]. Instead, utilizing ratios combining the PI of the MCA with that of the umbilical artery or uterine artery are more useful [98,99,100]. The MCA to umbilical artery resistance index is known as the cerebroplacental ratio and is a good predictor of adverse neonatal outcome [101,102], with a higher diagnostic accuracy in predicting IUGR than either measurement alone [98]. Measurement of the MCA to uterine artery PI ratio has also shown promising results of identifying unfavourable foetal outcomes [99,100]. With disturbance to placental blood flow, resistance is increased in the uterine artery [103] and, if the pregnancy is associated with growth restriction, resistance is decreased in the MCA [97]. Two studies have demonstrated that in preeclampsia, at or beyond 26 weeks’ gestation, a low MCA/uterine artery PI is associated with higher rates of unfavourable foetal outcomes including IUGR, caesarean sections, and preterm birth, and it provides a better prediction of outcome than umbilical artery Doppler [99,100]. Though these trials were both small, the findings were consistent and further research may confirm clinical utility.

In addition to surveillance methods currently recommended by national obstetric guidelines, studies have considered other measures to predict adverse foetal outcomes in preeclamptic pregnancies. Biochemical tests normally performed in preeclamptic pregnancies have limited use in predicting foetal complications; the degree of proteinuria and level of serum uric acid are poor predictors [104,105] and the presence of abnormal liver function tests can moderately predict foetal complications, but with poor sensitivity [106]. Preliminary data from small cohorts has found CRP levels and the nitric oxide inhibitor maternal plasma asymmetric dimethylarginine to be associated with foetal growth restriction in preeclamptic pregnancies [107,108], though evidence is not sufficient to suggest clinical use.

At present, due to a lack of evidence regarding foetal surveillance, there is disparity between guidelines. Additionally, the disease trajectory of preeclampsia can be unpredictable and can change rapidly, so the foetal surveillance techniques employed need to be altered appropriately based on the current maternal and foetal condition [43]. As a result, decisions regarding the modality and frequency for foetal assessment often rely on the practicing obstetrician and individual institutional guidelines.

### 4.3. Management

Interventions for the management and prevention of foetal complications of preeclampsia are limited. As preeclampsia is responsible for around 20%–30% of all preterm births [109], the management options available to optimize the condition of the foetus, such as administration of antenatal corticosteroids and magnesium sulphate infusions, are primarily aimed at preventing adverse outcomes associated with prematurity. Additionally, prevention of adverse foetal outcomes involves optimizing the timing of delivery. However, choosing the timing of delivery is not purely an intervention to reduce foetal complications, and requires a careful balance of the condition and gestation of the foetus and the condition of the mother. Managing preeclampsia also involves optimization of the maternal condition with antihypertensives, and magnesium sulphate if required [3], which may also provide benefits to the foetus.

Antenatal corticosteroids are recommended if a woman with preeclampsia is suspected to deliver prematurely (between 26 and 36 weeks’ gestation) within the next seven days [110]. A single course of corticosteroids (betamethasone or dexamethasone) is supported by robust evidence to reduce the risk of perinatal death and neonatal complications including respiratory distress syndrome (RDS), necrotizing enterocolitis, and intensive care admissions in pregnancies at risk of preterm birth [111]. While studies include women at risk of preterm birth regardless of the cause, subgroup analysis indicates there is no evidence to suggest any difference of effect in preterm birth as a result of hypertensive disease [111]. The optimal corticosteroid to use, mode of administration at which gestations steroids are effective, and whether repeat dosing is beneficial is less clear. A Cochrane review reported that it remains uncertain whether use of dexamethasone or betamethasone is preferred; one trial showed dexamethasone may reduce neonatal intraventricular haemorrhage rates more than betamethasone, though they were equivalent for other neonatal outcomes measured [112]. One small trial suggested intramuscular injection may be a more beneficial mode of delivery than oral, however, more evidence is needed to support this [112]. Though it is known that two doses of the corticosteroid are required, it is unclear whether an interval of 12 or 24 h between these doses is more effective [113]. In terms of gestation, most evidence for the administration of antenatal corticosteroids to minimize foetal and neonatal complications supports its use between 26 and 34 weeks’ gestation [111]. After 34 weeks’ gestation, evidence is less robust, with some trials showing no benefit [114]. Nevertheless, a subgroup analysis by gestation in a Cochrane review suggested that despite less evidence, there remains a clear clinical benefit of corticosteroids to reduce RDS after 34 weeks, suggesting any risks are outweighed by potential benefits [111]. Additionally, a meta-analysis has shown that if an elective caesarean is planned, antenatal corticosteroids may still reduce rates of RDS after 37 weeks’ gestation [115]. In spite of this, corticosteroids have been shown to cause long-term changes in the vasculature and glucose metabolism of the offspring [116], which needs to be considered on balance with the benefits at later gestations.

Several RCTs have assessed whether multiple courses of antenatal corticosteroids are beneficial if after the initial dose, a woman does not deliver within seven days and is still at risk of preterm birth. Results of three large RCTs are conflicting [117,118,119]. One trial of 982 women at less than 32 weeks of pregnancy who were at high risk of preterm birth saw them randomized seven or more days after initial corticosteroid treatment to weekly repeat dosing of betamethasone or placebo injections [117]. They reported a reduction in RDS and severe lung disease, but lower offspring weight and head circumferences at birth. These findings are consistent with another RCT of 1348 women between 28 and 35 weeks of pregnancy [119]. In contrast, another large RCT of 1858 women did not find any improvement in birth outcomes with repeat dosing of betamethasone, but also reported a detrimental effect on birth size [118]. All studies used intramuscular betamethasone of a similar dose, but the most noticeable difference in methodology was the frequency of repeat dosing, which was weekly, every ten days, and fortnightly in the three studies respectively. Two of the trials reported follow-up at two years, and one at five years [120,121,122]. In all of these long-term follow-up studies no differences were noticed in mortality, body weight, or neurodevelopmental disability [120,121,122]. Due to the lack of certainty in benefit and risks, at present guidelines do not recommend repeat courses of corticosteroids as routine care [110].

Magnesium sulphate is given as primary and secondary prophylaxis of seizures in women with preeclampsia regardless of gestation and is also recommended in planned or expected preterm delivery for its neuroprotective effects in the offspring. NICE recommends infusion within 24 h in women with preeclampsia between 24 and 30 weeks’ gestation, and should be considered in women up to 34 weeks’ gestation [3]. Magnesium sulphate appears safe for the foetus and meta-analysis has shown that administration antenatally can reduce the risk of cerebral palsy in the offspring [123]. However, the majority of large trials conducted have excluded women delivering preterm as a consequence of preeclampsia. Data from the largest trial that did include women with preeclampsia did not find any significant difference in neonatal morbidity [124], childhood death, or disability at 18 months in those whose mothers were given magnesium sulphate or placebo [125]. Although, there was a tendency to reduction in death and cerebral palsy with magnesium sulphate, this did not reach statistical significance [125]. Therefore, while magnesium sulphate is recommended for foetal neuroprotection in women with preeclampsia at risk of imminent delivery at less than 34 weeks’ gestation, the evidence supporting this recommendation largely comes from trials in which preeclampsia was excluded as a cause of preterm birth.

The only definitive treatment for preeclampsia is delivery. Optimal timing of delivery requires a careful balance of maternal and foetal risks, including the gestation of the foetus. Overall, indications for planned early delivery are usually maternal, however foetal complications such as abnormalities in foetal ultrasound or CTG monitoring may also result in the decision for early birth [3]. Adverse outcomes in the offspring, including perinatal mortality, are strongly linked to the gestational age at delivery. In general, from a foetal perspective, at early gestations continuation of the pregnancy is desirable in order to improve prognosis [126] unless there is severe placental dysfunction. Therefore, in the absence of other indications, recommendations regarding expectant or immediate management and the optimal timing of delivery differ depending on the gestational age at which preeclampsia is diagnosed. If the onset of preeclampsia is before 24 weeks’ gestation (pre-viable), continuing with the pregnancy may not be advisable, due to high maternal morbidity and mortality rates and a low chance of offspring survival [127]. From 24 to 34 weeks of pregnancy, delivery within 24–48 h is associated with increased risks of adverse events for the foetus and neonate. These include intraventricular haemorrhage, low birth weight, longer stays in intensive care, and increased requirement for respiratory support [128]. Therefore, in women with preeclampsia before 34 weeks’ gestation, delaying delivery if possible is likely to be beneficial for the offspring. Between 34 and 37 weeks’ gestation, immediate delivery appears beneficial for the mother, however, it can increase adverse neonatal outcomes such as RDS, especially if at less than 36 weeks’ gestation [129]. The decision to deliver the foetus immediately or to continue expectant management at these gestations can be difficult. In general, it is advised to continue expectant management unless there are indications that immediate delivery is required [3,129]. The largest trial to date assessing optimal time for delivery in preeclamptic pregnancies between 34 and 37 weeks’ gestation is currently underway and should provide further evidence [130]. After 37 weeks’ gestation, expectant management or interventional management does not appear to affect neonatal outcomes, however interventional management is beneficial for the mother and is therefore advised [131].

These recommendations according to gestation may not be applicable in the presence of maternal or foetal complications that require more immediate delivery. Foetal indications for delivery may include severe IUGR or evidence of worsening foetal compromise on surveillance modalities, indicating foetal hypoxia [3,43,84]. More specifically, this may include persistent reversed end-diastolic flow in umbilical artery Doppler velocimetry or a nonreassuring CTG, though no consensus exists regarding the most appropriate trigger for delivery [3,84]. If foetal compromise is present, delaying delivery may damage brain development due to prolonged foetal hypoxia, yet early delivery carries the risks associated with prematurity. One study designed to assess this randomised 548 pregnant women to early or delayed delivery. Women were between 24 and 36 weeks of pregnancy with signs of foetal compromise and uncertainty about the most appropriate time to deliver. No difference was seen in infant mortality prior to hospital discharge, or in death or disability at two years of age [132,133], leaving the optimal management unclear. In this study, 43% of the pregnancies were hypertensive, but further evidence is lacking regarding when to deliver in the setting of foetal compromise specifically in preeclamptic pregnancies.

Assessment of the ductus venosus flow may provide a promising method to predict perinatal outcome and appropriately time delivery in preeclamptic pregnancies complicated by IUGR. A prospective cohort study identified that in pregnancies complicated by early onset IUGR, an abnormally high ductus venosus pulsatility index can predict adverse foetal outcome more accurately and earlier than changes in foetal heart rate or by umbilical artery Doppler [134]. It may therefore be a more useful parameter for timing delivery. This has been further assessed in a recent multicentre randomised trial of pregnancies complicated by early onset IUGR, in which ∽50% were preeclamptic [135]. This study compared neonatal outcomes when three different antenatal monitoring strategies were used in order to time delivery: reduced foetal heart rate on CTG, early ductus venosus changes (as measured by high pulsatility index), or late ductus venosus changes (measured by the absence of an A wave). They found that by two years of age, neonates assigned to the cohort in which prediction of delivery was based on late ductus venosus changes had a significant reduction in neurodevelopmental impairment when corrected for prematurity. This suggests that a more conservative approach in timing delivery, by waiting for late ductus venosus changes, may improve perinatal outcomes.

Novel management options studied to reduce adverse foetal outcomes of preeclampsia include antithrombin, sildenafil, pravastatin, metformin, and plasma exchange. Antithrombin has been shown in a small trial to preserve foetal biophysical profile and weight gain in early onset preeclampsia (<32 weeks) [136]. Most evidence regarding sildenafil comes from studies on preeclamptic rat models in which foetal blood flow and outcomes have been improved with its use [137]. There has also been a small preliminary trial of sildenafil in women with severe IUGR, which has shown an improved foetal growth velocity and a trend towards improved perinatal survival. In animal models, pravastatin has been demonstrated to reduce IUGR, lower blood pressure, and improve the balance of angiogenic factors by promoting release of PlGF and suppressing sFlt-1 and soluble endoglin production [138]. Similarly, pravastatin administration in human umbilical vein endothelial cells (HUVECs), cytotrophoblasts, and placental tissue results in reduced markers of endothelial dysfunction [139]. Despite concerns of teratogenicity, pravastatin use in small studies has not been associated with adverse pregnancy outcomes including birth defects [138,140] and further studies are currently underway to validate use as a potential treatment. (StAmP trial: Statins to Ameliorate Early Onset Preeclampsia (ISRCTN23410175)). Metformin has also emerged as a potential treatment option for preeclampsia. In preclinical studies using primary human tissue, metformin reduced sflt-1 and endoglin secretion, improved features of endothelial dysfunction and angiogenesis, and enhanced vasodilation [141]. Due to use in diabetes, metformin is known to be safe in pregnancy, though RCTs evaluating potential use as a treatment for preeclampsia are lacking. Plasma apheresis to facilitate removal of antiangiogenic protein sFlt-1 has also been evaluated as a treatment method. Small pilot studies of apheresis in women with very preterm preeclampsia have shown a reduction in circulating sFlt-1 without apparent adverse maternal or foetal events, and a potential prolongation of pregnancy, warranting larger trials to confirm its utility [142,143]. These options do not have sufficient evidence for their use in clinical practice. As such, clinicians rely on monitoring and timely delivery, with the consideration of antenatal corticosteroids and/or magnesium sulphate depending on gestation, as options for improving foetal and neonatal outcomes in preeclampsia.

## 5. Long-Term Impact on the Offspring

There is growing evidence that there are long-term cardiovascular sequelae in the mother following hypertensive pregnancies [144,145,146] and in the offspring (Figure 1) from in utero exposure to hypertensive disorders of pregnancy, which are independent from other coexisting pregnancy complications. Meta-analysis using data from over 45,000 individuals reported a 2.39 mmHg higher systolic, and 1.35 mmHg higher diastolic blood pressure, in children and young adults born to preeclamptic pregnancies [7]. If this difference tracked into adult life, it would be associated with an 8% increased risk of mortality from ischemic heart disease and a 12% increased risk of stroke [7].

These findings are supported by a 20 year prospective follow-up birth cohort study of 2868 young adults, which reported that the clinical incidence of hypertension is increased in those exposed to hypertensive disorders of pregnancy in utero. These young adults were 2.5 times more likely to have global lifetime risk (QRISK) scores above the 75th centile, and 30% of 20-year-olds with high blood pressure were born following a hypertensive pregnancy [147]. These increases in blood pressure have been shown to track into later life, with one study showing that offspring of preeclamptic pregnancies were more likely to be prescribed antihypertensive medication by 50 years of age [148]. Indeed, a 60 year follow-up of the Helsinki birth cohort also demonstrated that individuals born following severe preeclampsia have a 1.5 relative risk of hypertension [149].

The timing of onset of preeclampsia also appears to be an important determinant in the development of later-life hypertension for the offspring. In one study comparing offspring born to normotensive pregnancies, early onset (<34 weeks’ gestation), and late onset (≥34 weeks’ gestation) preeclamptic pregnancies, blood pressure differences were seen exclusively in the offspring of early onset preeclamptic pregnancies at six and 13 years of age [150]. Those born from mothers with early onset preeclampsia were found to have a 6 mmHg increase in peripheral and central systolic blood pressure, a noticeably greater increase than in studies without discrimination between early or late onset disease [7]. As earlier onset preeclampsia also commonly occurs with IUGR and prematurity, it can be difficult to separate the effects of these on the development of later hypertension. Nevertheless, individuals born preterm to preeclamptic pregnancies have been shown to exhibit specific vascular differences [151]. Moreover, recent research involving 15,000 young adults has described that the siblings of offspring born from preeclamptic pregnancies are also at higher risk of hypertension later in life, even if the mother was not hypertensive in that pregnancy [152]. This suggests that the traditional explanation of in utero stress and developmental reprogramming may not be sufficient to explain the risk of hypertension in those born to a mother with preeclampsia, and suggests a need to explore genetic and epigenetic factors, as well as maternal cardiovascular remodelling, to explain differences in the cardiovascular phenotypes of the offspring [153].

There is a growing body of evidence to suggest that the offspring of preeclamptic pregnancies have a distinct vascular phenotype, which may mediate the increased risk of hypertension [154]. Alterations in vascular function, including abnormal endothelial dilation, and changes in vascular structure, such as arterial thickening, have been found in animal models and human studies on offspring of preeclamptic pregnancies [154]. Lazdam et al. demonstrated that preterm-born young adults have impaired flow mediated endothelial responses only if they were exposed to a hypertensive pregnancy [151]. An increased carotid intima-media thickness was also found in these individuals [151], suggesting an early atherogenic phenotype consistent with aortic arterial thickening. A similar phenotype has also been shown in preeclamptic offspring at birth [155]. Endothelial dysfunction has also been demonstrated in childhood [156] and adolescence [157], although the results are not consistent across studies [158] and further research is needed to better understand between population heterogeneity.

While the precise mechanisms are yet to be fully elucidated, potential mediators for the abnormalities in vascular development in the offspring of preeclamptic pregnancies include derangement of maternal angiogenic factors, inherited polymorphisms, epigenetic factors, and altered microRNA expression [154,159,160]. Yu et al. [159] analysed the maternal angiogenic profile alongside assessments of neonatal endothelial cells at birth, and vascular density in the offspring at three months postpartum in hypertensive and normotensive pregnancies. At birth, HUVECs from the offspring of hypertensive pregnancies exhibited a lower vasculogenic capacity compared to those of normotensive pregnancies. This correlated with the in vivo microvascular findings in which the offspring born to hypertensive pregnancies exhibited an approximately two-fold reduction in microvascular density in the early postnatal period. This disruption of microvascular development was proportional to peripartum levels of maternal sFlt-1, implicating a role for maternal angiogenic factors in the abnormal development of the foetal vasculature in hypertensive pregnancies [159]. The neonatal microRNA profile also appears to play a role in mediating endothelial changes in the offspring of hypertensive pregnancies [160]. In particular, a specific microRNA involved in endothelial gene regulation, miR-146a, was shown to differ significantly between offspring of normotensive and hypertensive pregnancies, with direct correlations to HUVEC proliferation capacity and tube formation. This elevation in miR-146a expression in HUVECs from hypertensive pregnancies at birth identified cells with reduced ability for in vitro vascular tube formation, which was rescued by miR-146a inhibition. In contrast, miR-146a overexpression significantly reduced vascular tube formation in HUVECs from normotensive pregnancies.

In addition to these vascular changes, a recent study has also discovered differences in the cardiac structure in adolescent offspring exposed to hypertensive disorders of pregnancy, with evidence of adverse cardiac remodelling [161]. Exposure to maternal hypertension was associated with a greater left ventricular wall thickness compared to controls, while those exposed to preeclampsia also demonstrated a reduced left ventricular end-diastolic volume. Similarly, in a study by Lewandowski et al. in preterm-born young adults, preterm offspring of hypertensive pregnancies were shown to have an additional reduction in left ventricular global peak systolic longitudinal strain compared to preterm-born young adults born to normotensive pregnancies [162]. Whether these changes are present earlier in life and whether they are of relevance to future cardiovascular disease risk in these populations will be of interest. Alterations in cardiac autonomic function in the offspring, another predictor of cardiac sequalae, as well as the greatest cardiac remodelling changes in early postnatal life, appear to be associated with prematurity but not hypertensive pregnancies per se [163,164].

## 6. Conclusions

Hypertensive disorders of pregnancy now affect around one in 10 pregnancies worldwide [1]. This incidence is continuing to increase worldwide and will continue to do so not only in westernized countries as mothers conceive increasingly later in life, but also in continents where the vast majority of our population now resides. This includes India, Asia, Africa, and South America, where changes in socioeconomic status are resulting in a shift towards conception at an older age, coupled to increasing obesity rates worldwide. In the face of this, advances in screening, detection, and diagnosis as well as the antenatal, perinatal, and postnatal management of preeclampsia are required for the mother and their offspring.

For women at high risk, home blood pressure self-monitoring to improve the detection of hypertension in pregnancy may be warranted. This could be particularly effective in areas of the world where access to regular antenatal clinics and midwives remains is limited. Telemonitored blood pressure offers further potential for earlier diagnosis in remote areas, which is currently being trialled as part of the CRADLE programs [165,166]. At present, aspirin is the only therapy with robust evidence to reduce the risk of preeclampsia in high-risk women [32]. Current recommendations advise low dose (75–150 mg) aspirin as prophylaxis from 12 weeks’ gestation until delivery [3]. Other interventions including nutritional supplements, pharmacological agents, and dietary and lifestyle interventions have been investigated for protective effects against preeclampsia with varying efficacy [33,34,35,36,37,38,39,40,41,42,43,44,45,46,47,48,49,50,51,52,53,54,55,56,57,58,59,60,61,62]. At present, interventions to reduce the risk of early foetal complications of preeclampsia remain limited and include administration of antenatal corticosteroids and magnesium sulphate infusions, which are primarily aimed at preventing adverse outcomes associated with prematurity. Further research in this field is needed to better understand the potential maternal and offspring benefits of dietary, lifestyle, and home-monitoring interventions for the pre- and postdelivery management of preeclampsia.

## Figures and Tables

**Figure 1 jcm-08-01625-f001:**
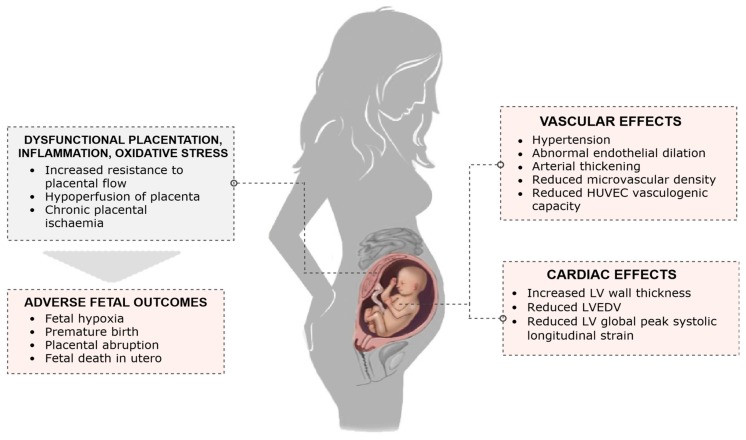
Effects of preeclampsia on the foetus and offspring. HUVEC stands for human umbilical vein endothelial cells; LV, left ventricle; LVEDV, left ventricular end-diastolic volume.

**Table 1 jcm-08-01625-t001:** Diagnostic criteria for preeclampsia.

Preeclampsia is Defined as Gestational Hypertension Associated with New-Onset Maternal or Uteroplacental Dysfunction at or after 20 Weeks’ Gestation	
**Gestational hypertension**	
Systolic blood pressure ≥ 140 and/or diastolic blood pressure ≥ 90	
Blood pressure should be repeated to confirm true hypertension	
A liquid crystal sphygmomanometer should be used with appropriate size cuff. Or, if unavailable an appropriately calibrated automated device.	
**Accompanied by at ≥1 of the following new-onset conditions:**	
Proteinuria	Initial assessment with automated dipstick urinalysis. If unavailable, visual analysis can be used.
	If dipstick is positive (≥1+), confirmed with spot urine. Abnormal if P:Cr ≥ 30 mg/mmol or A:Cr ≥ 8 mg/mmol
Renal complications	Acute Kidney Injury (creatinine ≥ 90 umol/L)
Liver complications	Elevated transaminases, with or without right upper quadrant of epigastric abdominal pain
Neurological complications	Eclampsia, altered mental status, blindness, stroke, clonus, severe and persistent visual scotomata
Haematological complications	Thrombocytopenia (platelet count < 150000/µL, disseminated intravascular coagulation, haemolysis)
Uteroplacental dysfunction	Foetal growth restriction, abnormal umbilical artery Doppler wave form analysis, stillbirth

**Table 2 jcm-08-01625-t002:** Routine recommendations for foetal surveillance in preeclampsia without severe features.

Guideline	Cardiotocograph	Biophysical Profile	Amniotic Fluid Volume	Umbilical Artery Doppler	Ultrasound for Foetal Growth
NICE (United Kingdom) ^†^ [3]	At diagnosis. If normal, do not routinely repeat unless indicated.	Not recommended	At diagnosis and every two weeks.	At diagnosis and every two weeks.	At diagnosis and every two weeks.
SOMANZ (Australia and New Zealand) * [43]	Twice weekly or more frequently if indicated.	Not recommended	At diagnosis and every two to three weeks.	At diagnosis and every two to three weeks.	At diagnosis and every two to three weeks.
ACOG (United States of America) [44,84]	At diagnosis, then twice weekly.	If CTG is non-reactive.	At diagnosis, then at least once weekly.	Adjunct if there is evidence of foetal growth restriction.	At diagnosis and every three to four weeks.
SOGC (Canada) [64]	Recommended, however timing not specified.	Not recommended	Recommended, however timing not specified.	Recommended, however timing not specified.	Recommended. Timing not specified.

NICE, National Institute for Health and Care Excellence; SOMANZ, Society of Obstetric Medicine of Australia and New Zealand; ACOG, The American College of Obstetricians and Gynecologists; and SOGC, Society of Obstetricians and Gynaecologists of Canada. ^†^ Subsequent surveillance and monitoring depending on scan findings. * These recommendations are suggested as a commonly used guideline. Individual units are advised to develop their own protocols.

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
