# Peer review of "Preeclampsia: Risk Factors, Diagnosis, Management, and the Cardiovascular Impact on the Offspring"

_jcm, 2019, doi:10.3390/jcm8101625_

Round 1

Reviewer 1 Report

This is a very detailed and well-written review regarding preeclampsia.

I have only a few comments and recommendations:

Abstract: Oligohydramnios is mostly a consequence or sort of manifestation of placental insufficiency due to preeclampsia. I would not describe oligohydramnios as adverse fetal outcome.

Line 41: preterm birth in preeclampsia is mostly iatrogenic and not spontaneous.

Line 115-118: although it is a known fact, that nicotine abuse seems to reduce the risk of preeclampsia, I would delete the following sentence: "However, this is not useful for clinical practice, given the well-documented detrimental effects of
 smoking on other adverse outcomes of pregnancy". "It is not useful.." is a little to succinct, given the fact, that nicotine abuse in pregnancy might harm the fetus similar to preeclampsia itself.

Line 155-158: The angiogenic dysbalance of sflt-1 and plgf seems to provoke the multisystem endothelial dysfunction leading to the specific clinical features of preeclampsia. The authors should go into more detail regarding sflt-1 and plgf.

Management: Line 350:  The authors should mention prolongation of pregnancy in cases of early onset preeclampsia

Line 404-406: other management options include pravastatin, metformin and plasma exchange.

Reviewer 2 Report

Major comments:

Well written review of the clinical diagnosis and management of preeclampsia, as well as the cardiovascular effects on the offspring. My only major comment is that it strikes me as a little odd to go into so much detail on the cardiovascular effects on the offspring, without touching more on the cardiovascular effects on the mother, which are mentioned but not discussed in any detail. If space allows, I would suggest adding another short section discussing these effects, as they are of great clinical importance.

Minor edits:

Abstract:

Line 14: Rephrase this, it sounds like you are saying 3-5% of the 10% (rather than of all pregnancies) are preeclamptic pregnancies.

Line 15: Replace "it" with "preeclampsia"

Introduction:

Line 35: I believe that the umbrella definition "hypertensive disorders of pregnancy" also includes chronic hypertension, chronic hypertension with superimposed preeclampsia, and eclampsia.

Line 37: instead of "the former" say "the mother"

Line 44: misplaced modifier - rearrange the sentence

Risk factors

Lines 115-118: Any speculation as to why? Is this true even after adjusting for obesity?

General: What about risk of recurrent preeclampsia after preeclampsia in a prior pregnancy?

Diagnosis: Interesting that neurological dysfunction is considered a defining feature of preeclampsia, but there is no guidance on how this should be diagnosed (via neuroimaging, or neurological consultation, or symptoms alone?)

General: I suggest mentioning in the first paragraph that preeclampsia can also first present and be diagnosed postpartum. In addition, it is worth defining eclampsia here.

Line 223: "in preeclamptic pregnancies"

Management

Line 288: Isn't magnesium given more for the purpose of seizure prophylaxis (to prevent eclampsia) rather than to avoid preterm delivery?

Long term impact

Line 454: Increased intima-media thickness in which artery specifically? (Carotid?)

Reviewer 3 Report

The review submitted by Fox and colleagues is an excellent overview of the state of current knowledge about preeclampsia, including immediate care for the patient, and long-term effects on the offspring. The authors have thoroughly scanned the relevant literature in this topic and produced a well-written, comprehensive review.

There are only minor issues to address in the current manuscript. Since the target audience for this review will likely comprise many clinicians in this field, it may be valuable to have a table or flow diagram describing the most current and validated diagnostic criteria for preeclampsia, i.e. a summary of section 3 of the manuscript. This may be particularly valuable since application of these criteria are quite inconsistent across various jurisdictions, and may help clear up confusion. Additionally a figure describing the impacts of preeclampsia on the maternal and fetal systems (such as cardiovascular, neurological, etc.) may help to add some more appeal to the manuscript.

Finally the manuscript lacks description of mechanistic studies in regards to the origin, pathogenesis, or treatment of preeclampsia. Recognizing that there is a paucity of human data in these regards, the authors might consider adding data derived from animal studies, as was done in the last paragraph of section 4.3.

Round 2

Reviewer 1 Report

All my questions and concerns have satisfactory been changed over.